# Combined Action of Anti-MUC1 Monoclonal Antibody and Pyrazole-Platinum(II) Complexes Reveals Higher Effectiveness towards Apoptotic Response in Comparison with Monotherapy in AGS Gastric Cancer Cells

**DOI:** 10.3390/pharmaceutics13070968

**Published:** 2021-06-26

**Authors:** Katarzyna Supruniuk, Robert Czarnomysy, Anna Muszyńska, Iwona Radziejewska

**Affiliations:** 1Department of Medical Chemistry, Medical University of Białystok, ul. Mickiewicza 2a, 15-222 Białystok, Poland; katarzyna.supruniuk@umb.edu.pl; 2Department of Synthesis and Technology of Drugs, Medical University of Białystok, ul. Kilińskiego 1, 15-089 Białystok, Poland; robert.czarnomysy@umb.edu.pl (R.C.); anna.muszynska@umb.edu.pl (A.M.)

**Keywords:** anti-MUC1, apoptosis, carbohydrate antigens, gastric cancer, platinum complexes

## Abstract

MUC1 mucin is a transmembrane glycoprotein aberrantly overexpressed and underglycosylated in most epithelium origin cancers. Combining chemotherapeutics with monoclonal antibodies toward cancer-related antigens is one of the new strategies in cancer therapies. In this study, we assessed the effectiveness of 10 μM cisplatin (cisPt), two pyrazole-platinum(II) complexes (PtPz4 and PtPz6), and 5 μg/mL anti-MUC1 used as monotherapy, as well as cisplatin and its derivatives combined with mAb on apoptotic response and specific cancer-related sugar antigens in AGS gastric cancer cells. Flow cytometry, RT-PCR, Western blotting, and ELISA tests were applied to determine the influence of examined compounds on analyzed factors. PtPz6 combined with anti-MUC1 revealed the strongest apoptotic response compared to control and monotherapy. The combined action of both cisPt derivatives and anti-MUC1 was more effective than monotherapy in relation to *Bad, Bcl-xL, Bcl-2, caspase-9, caspase-3*, as well as pro- and cleaved *caspase-3* protein, and T, sialyl Tn sugar antigens in cell lysates, and Tn, T, sialyl Tn, sialyl T antigens in culture medium. Additionally, PtPz4 administrated with mAb was revealed to be more potent than used alone with regard to *Bax* protein and *Bid* expression, and PtPz6 used in complex with anti-MUC1 revealed more efficient action towards *Akt* and sialyl T antigen expression. These data indicate the rationality of the potential application of combined treatment of anti-MUC1 and cisPt derivatives in gastric cancer therapy.

## 1. Introduction

Gastric cancer (GC) has been stated as one of the leading causes of tumor-related deaths globally [1]. Most of the GC patients are diagnosed with late stages, with poor prognosis and limited possibilities of effective therapies. Cisplatin, a common chemotherapeutic, is widely applied in the treatment of various human cancers [2,3]. Its action in cancer cells is based on the potential to crosslink with the DNA purine bases, attenuating DNA repair, leading to DNA damage, and consequently inducing apoptosis. While cisplatin is effective, its usage can be limited by its severe side effects, as neurotoxicity, anaphylaxis, cytopenias, and others. Furthermore, resistance to this drug can also reduce its application in many tumors [4,5]. Therefore, in the last years, numerous platinum-based compounds were synthesized with the goal of reducing adverse effects and overcoming drug resistance [6]. Recent studies on breast cancer cells revealed that pyrazole-platinum(II) complexes might induce cell cycle arrest and apoptosis, therefore constituting potential anti-cancer agents [7]. Apart from that, there have been attempts to apply combination therapies of cisplatin with other drugs.

The human mucin 1 (MUC1) is a high-molecular-weight membrane-bound glycoprotein overexpressed in most human epithelial cancers, including gastric cancer [8]. It consists of two subunits: intra- (C-terminal; MUC1-C) and extracellular (N-terminal; MUC1-N) bound by non-covalent interactions. MUC1 normally occurs on the apical surface of the cell membrane whereas, in cancer, overexpression and loss of polarity result in mucin presence circumferentially over the entire surface of the cell [8,9]. The MUC1-N includes a variable number of tandem repeats (VNTR) region composed of a sequence of 20 amino acids repeated 20–200 times, extensively modified by O-glycosylation. Glycoproteins participate in communication with other cells and matrix components [9,10]. The MUC1-C contains an extracellular domain, transmembrane domain, and a cytoplasmic tail (CT). Through MUC1-CT phosphorylation and association with various proteins, mucin can modulate intracellular signaling pathways connected with oncogenic transformation [8,11,12]. MUC1 has been reported to play a role in cancer development by suppression of cell death and promotion of metastasis [13,14]. This mucin seems to be attractive as a target antigen due to its overexpression in many carcinomas in comparison to normal tissues. Therefore antibodies towards tumor-associated MUC1 are more likely to bind to the antigen on the tumor cells and not on normal cells. Additionally, MUC1 underglycosylation and specific localization over the entire surface of the cell makes the molecule easily available for antibodies. Over the past 30 years, MUC1 mucin has remained a potential diagnostic marker and therapeutic agent [13]. There are anti-MUC1 based therapies developed against gastric cancer; some of them are in clinical trials or currently under pre-clinical studies [15,16]. Moreover, there are attempts to apply therapies based on the combined action of anti-MUC1 and cisplatin complexes. Gornowicz et al. revealed the anti-cancer action of platinum(II) complexes combined with anti-MUC1 in MDA-MB-231 and MCF-7 breast cancer cells [17,18].

The promising results of research on breast cancer cells raise the question of whether, in other cancers, the combination of an anti-MUC1 antibody and a chemotherapeutic agent, pyrazole-berenil complexes of platinum(II), will also give positive results in the stimulation of apoptosis in transformed cells. Therefore, in this study, we investigated the effect of anti-MUC1 and dinuclear pyrazole-platinum(II) complexes, [Pt_2_(pyrazole)_4_(berenil)_2_]·4HCl·2H_2_O (PtPz4) and [Pt_2_(N-ethylpyrazole)_4_(berenil)_2_]·4HCl·2H_2_O (PtPz6), on apoptotic response in gastric cancer AGS cells.

## 2. Materials and Methods

### 2.1. Cell Culture

Human gastric adenocarcinoma cells CRL-1739 (AGS) were purchased from American Type Culture Collection (Manassas, VA, USA) and cultured in F-12 medium (Gibco, Waltham, MA, USA), 5% CO_2_ at 37 °C. The medium was supplemented with 10% Fetal Bovine Serum (FBS) (Gibco, Waltham, MA, USA), 100 U/mL penicillin, and 100 μg/mL streptomycin (Sigma, St. Louis, MO, USA). Then, cells were seeded into six-well plates in 1 mL of growth medium, and after reaching about 70% of confluency, were used for the assays. The platinum complexes used in the study [Pt_2_(pyrazole)_4_(berenil)_2_]·4HCl·2H_2_O (PtPz4) and [Pt_2_(N-methylpyrazole)_4_(berenil)_2_]·4HCl·2H_2_O (PtPz6) (Figure 1) were synthesized using previously described methods [7].

Cisplatin (Sigma, St. Louis, MO, USA), PtPz4, and PtPz6 were dissolved in dimethyl sulfoxide (DMSO; Sigma, St. Louis, MO, USA) (concentration of the compounds in stock solutions was 10 mM) and diluted with supplemented medium to obtain a final concentration of 10 μM. Cells were incubated for 24 h with FBS-free medium alone (control), medium containing PtPz4, PtPz6, cisplatin (cisPt), and monoclonal anti-MUC1 antibody (5 μg/mL). For combined treatment, AGS cancer cells were pretreated with anti-MUC1 mAb 5 h before platinum and platinum complexes addition. After the washing step (Phosphate Buffered Saline (PBS); pH 7.4), cell lysis was performed using Radio-Immunoprecipitation Assay (RIPA) buffer (Sigma, St. Louis, MO, USA) blended with protease inhibitors (Protease Inhibitor Coctail; Sigma, St. Louis, MO, USA) (dilution 1:200 in RIPA buffer) for 20 min at 4 °C. Upon intensive vortexing and centrifugation at 1000× *g* for 5 min at 4 °C, the cell lysates and culture media were collected, frozen at −70 °C, and then used for Western blot analyses and ELISA tests. To assess the protein concentration in the cell lysates, a BCA Protein Assay Kit (Pierce, Rockford, IL, USA) was used. For real-time PCR assay, the monolayers were washed three times with 10 mM PBS. Then the cells were collected into sterile plastic tubes and disrupted by a sonicator (Sonics Vibra cell; Sonics & Materials, Leicestershire, UK) (10 W, three times for 15 s on ice). For RNA isolation, aliquots of the homogenate were used.

### 2.2. Cell Viability Assay

The measurements of the viability of the cultured cells were performed in accordance with Carmichael et al. [19], with 3-(4,5-dimethylthiazole-2-yl)-2,5-diphenyltetrazolium bromide (MTT) (Sigma, St. Louis, MO, USA). Cells were cultured on six-well plates, and after obtaining 70% of confluency, they were incubated for 24 h with 2–100 μM concentrations of PtPz4, PtPz6, and cisPt. After that, the cells were incubated for 4 h in MTT solution (0.5 mg MTT/mL of PBS) in a 5% CO_2_ atmosphere at 37 °C. The absorbance of the transformed dye in living cells was measured at 570 nm. The viability of AGS cells in the presence of studied compounds was calculated as a percentage of control cells (with 100% cell viability). Cell viability determination for anti-MUC1 was published previously in the work of Radziejewska et al. [20].

### 2.3. Annexin V Binding Assessment

To evaluate the cell death induced by the examined compounds, Apoptosis Detection Kit II (BD Pharmingen, San Diego, CA, USA) was used according to the manufacturer’s instructions. Shortly after cells were trypsinized, they were resuspended in F-12 medium and then in binding buffer. Subsequently, they were stained with FITC Annexin V and propidium iodide (PI) at RT (room temperature) in the dark (15 min). Annexin V bound with high affinity to phosphatidylserine was used to identify cells in the early stages of apoptosis, and PI was applied to determine late apoptotic and dead cells. Cells cultured in a drug-free medium were used as a negative control. The obtained results were evaluated on FACSCanto II cytometer (BD Biosciences, San Jose, CA, USA) with FACSDiva software (BD Biosciences, San Jose, CA, USA). Cells incubated with 3% formaldehyde in buffer for 30 min on ice were considered as positive control and applied to find the optimal parameter settings.

### 2.4. RNA Isolation and Quantitative Real-Time PCR

Total RNA was extracted using Total RNA Mini Plus Concentrator (A&A Biotechnology, Gdynia, Poland) in accordance with the manufacturer’s guidelines. Isolated RNA concentration was evaluated using a NanoDrop 2000 spectrophotometer (Thermo Scientific, Waltham, MA, USA). Equal amounts (1 μg) of total RNA were used as the template for complementary DNA (cDNA) synthesis using SensiFAST^TM^ cDNA Synthesis Kit (Bioline, London, UK). The reaction mixture (20 μL) containing 4 μL of 5× TransAmp Buffer, 1 μL of Reverse Transcriptase, RNA template, and DEPC-treated water was incubated for 10 min at 25 °C, 30 min at 45 °C and 5 min at 70 °C. Real-time PCR was performed with CFX96 detection system (Bio-Rad, Hercules, CA, USA) using SensiFAST^TM^ SYBR Kit (Bioline, London, UK). 10 μL of reaction mixtures contained SensiFAST SYBR No-ROX Mix (5 μL), 3-times diluted cDNA template (2 μL), 0.4 μL (10 μM) of each target-specific primer purchased from Genomed (Warsaw, Poland) and DEPC-treated water. Primers for the experiment (forward and reverse) are listed in Table 1. Glyceraldehyde-3-phosphate dehydrogenase (GAPDH) was used as a control gene. The cDNA amplification conditions were as follows: 95 °C for 2 min for DNA polymerase activation and 40 cycles consisted of 95 °C for 10 s (denaturation), 60 °C for 15 s (annealing), and 72 °C for 20 s (elongation). Then, the melting curve was run to evaluate the specificity of each amplification. To confirm single product formation, melting point analysis and agarose gel electrophoresis were performed. Water, instead of mRNA samples, was used as a negative control. Each sample was analyzed in triplicate. Relative changes in mRNA expression were normalized with GAPDH and calculated using the delta–delta cycle threshold (ΔΔCt) method.

### 2.5. Western Blot Analysis

The samples prepared for electrophoresis were mixed with probe sample buffer (4:1) containing 2.5% sodium dodecyl sulfate (SDS) (Sigma, St. Louis, MO, USA). The same amount of proteins (20 μg) of cell lysates were separated on 7.5–13% polyacrylamide gels (concentration of the gel depending on the molecular weight of tested proteins) and immunoblotted on an Immobilon-P transfer membrane (Millipore, Bedford, MA, USA) according to Towbin et al. [21]. Next, the membranes were blocked with 5% non-fat milk in Tris-buffered saline (50 mM Tris-HCl, pH 7.5, 150 mM NaCl) supplemented with 0.05% Tween 20 (Sigma, St. Louis, MO, USA) (TBS-T) for 1 h at RT. Then, the membranes were washed three times (with TBS-T) and incubated overnight at 4 °C with specific monoclonal antibodies (listed in Table 2) diluted 1:600 in TBS-T containing 3% bovine serum albumin (BSA). To detect immunoreactive complexes, proper secondary horseradish peroxidase-conjugated antibodies diluted 1:2500 in TBS-T containing 5% non-fat milk were applied. To eliminate non-specific bindings, TBS-T buffer instead of monoclonal antibodies was used as a negative control. In certain analyses, membranes were stripped using a Restore Western Blot Stripping Buffer (Thermo Fisher Scientific, Waltham, MA, USA) and reused. Bound antibodies were visualized by enhanced chemiluminescence with Westar Supernova, a chemiluminescent substrate for Western blotting (Cyangen, Bologna, Italy). Band intensity was quantified by densitometry with an imaging densitometer (G:BOX, Syngene, Cambridge, UK) and normalized for β-actin.

### 2.6. ELISA Tests

To assess the relative level of Tn, T, sialyl Tn and sialyl T carbohydrate antigens, as well as MUC1 in cell lysates and culture media, ELISA tests were applied. To determine carbohydrate antigens, the following biotinylated lectins (Vector, Burlingame, CA, USA) at concentrations 5 μg/mL were used: VVA (from *Vicia villosa*) with affinity to Tn antigen (GalNAcα1-O-Ser/Thr), PNA (from *Arachis hypogaea* (peanut)) with affinity to T antigen (Galβ1-3GalNAcα1-O-Ser/Thr), SNA (from *Sambucus nigra*) with affinity to sialyl Tn (SAα2-6Gal/GalNAc), and MAAII (from *Maackia amurensis*) with binding preference to sialyl T (SAα2-3Gal). To detect MUC1 mucin, the anti-MUC1 monoclonal antibody (Table 2) was applied. 50 μL of cell lysates or culture medium (100 μg protein/mL) were coated on microtiter plates (NUNC F96; Maxisorp, Roskilde, Denmark) and incubated overnight (RT). Then, the wells were blocked 1 h at RT (with 100 μL of blocking ELISA reagent; Roche Diagnostics, Mannheim, Germany), washed three times (with 100 μL of washing buffer PBS, 0.05% Tween), and incubated with lectins or anti-MUC1 mAb 2 h at RT. In the next step, the plates were incubated 1 h at RT with 100 μL of horseradish peroxidase avidin D (Vector, Burlingame, CA, USA) for carbohydrate antigens detection or with secondary horseradish peroxidase-conjugated anti-mouse IgG antibody for MUC1 detection. The colored reaction was developed using 100 μL of ABTS (2,2′-azino-bis(3-ethylbenzthiazoline-6-sulfonic acid) (Sigma, St. Louis, MO, USA). Absorbance at a wavelength of 405 nm was read after 20–40 min. The samples were analyzed in triplicate. A 1% BSA solution instead of the samples and washing buffer instead of lectins and primary antibody were used as negative controls.

### 2.7. Statistical Analysis

The results are presented as mean ± standard deviation (SD) from at least 3 independent determinations. Statistical analyses were carried out using the Statistica package (StatSoft, Tulsa, OK, USA). To determine statistical differences, one-way ANOVA followed by Duncan’s multiple range post hock test was applied; *p* ˂ 0.05 was considered statistically significant.

## 3. Results

### 3.1. Cytotoxic Effects of PtPz4, PtPz6, cisPt, and Anti-MUC1

To assess the effects of PtPz4, PtPz6, and cisplatin on the viability of gastric cancer cells CRL-1739, a MTT assay was applied (Figure 2). The IC50 values for PtPz4 was 14 μM, for PtPz6 24.5 μM, for cisPt > 100 μM after 24 h of incubation. In all experiments performed in the study, we used 10 μM concentrations of PtPz4, PtPz6, and cisPt. IC50 for anti-MUC1was determined previously as 7 μg/mL [20]. In the present study, mAb with 5 μg/mL was applied.

### 3.2. Inhibitory Effect of Anti-MUC1 Monoclonal Antibody on MUC1

RT-PCR and ELISA tests were used to assess the effect of anti-MUC1 mAb action on MUC1 mucin expression in gastric cancer cells. Following 24 h of incubation of gastric cancer cells with monoclonal antibody (5 μg/mL) we observed a significant decrease in MUC1 mRNA expression compared to control (Figure 3A). The inhibition of the mucin expression was also observed in cell lysates (Figure 3B) and culture medium (Figure 3C), with the highest significant decrease in MUC1 released to the culture medium.

### 3.3. The Effect of PtPz4, PtPz6, cisPt, and Anti-MUC1 on the MUC1 Cytoplasmic Domain

Western blotting was applied to investigate the effects of examined compounds and mAb on the MUC1 cytoplasmic tail expression. The cells were incubated for 24 h with PtPz4, PtPz6, cisplatin, anti-MUC1 used in monotherapy and combined (PtPz4 + anti-MUC1, PtPz6 + anti-MUC1, cisPt + anti-MUC1). We found that cisplatin used alone and in combination with anti-MUC1 significantly increased mucin cytoplasmic domain expression (Figure 4). After combined cisPt and anti-MUC1 treatment lower stimulating effect was observed. Anti-MUC1 used in monotherapy and in combination with PtPz4 and PtPz6 reduced MUC1 cytoplasmic tail expression.

### 3.4. Impact of PtPz4, PtPz6, cisPt, and Anti-MUC1 on Apoptosis

Annexin-V and PI staining was used to assess the influence of examined compounds on apoptosis in gastric cancer cells (Figure 5). Annexin V allowed establishing cells in the early stages of the programmed cell death, while propidium iodide identified late apoptotic and dead cells. Analysis revealed that both platinum complexes used in the study did not significantly induce an apoptotic response in comparison to control. In cisplatin-treated cells, there were 16.9% apoptotic cells vs. 10.6% in control cells. After incubation with anti-MUC1, 15.14% of apoptotic cells were observed. The strongest pro-apoptotic effect on gastric cancer cells was revealed by combined treatment of PtPz6 and anti-MUC1 (26.8% of apoptotic cells). PtPz4 and cisPt combined with mAb also exhibited pro-apoptotic potential; there were 18 and 20.05% of apoptotic cells, respectively.

### 3.5. The Effect of PtPz4, PtPz6, cisPt, and Anti-MUC1 on NF-κB mRNA

NF-κB signaling is said to be critical in cancer development and can be deregulated in gastric cancer. RT-PCR determinations revealed that PtPz6, cisPt, and anti-MUC1 used alone as well as cisPt administrated with mAb significantly decreased NF-κB mRNA in comparison to untreated control (Figure 6). However, such inhibitory effects were not observed after monotherapy with PtPz4 and after combined action of anti-MUC1, PtPz4, and PtPz6.

### 3.6. The Effect of PtPz4, PtPz6, cisPt, and Anti-MUC1 on AKT mRNA

*Akt* controls the survival of the cells and inhibits apoptosis by phosphorylating specific components participating in the cell death program. RT-PCR results showed that only PtPz6 monotherapy and this compound combined with anti-MUC1 significantly suppressed *Akt* mRNA expression in comparison to control (Figure 7). There was no statistically significant impact of other tested variants of therapy on the *Akt* signaling pathway.

### 3.7. The Effect of PtPz4, PtPz6, cisPt, and Anti-MUC1 on Pro-Apoptotic Factors

To investigate if examined compounds initiate apoptosis by activation of pro-apoptotic factors, *Bax* mRNA and *Bax* protein, as well as *Bad*, Bid, and *Bim* mRNAs were determined. We found that *Bax* gene expression was significantly suppressed by PtPz6 monotherapy, and the rest of the examined compounds used in monotherapy and combined with anti-MUC1 had no significant impact on *Bax* mRNA. These results were not correlated with *Bax* protein expression, analyzed by Western blotting. We found that PtPz6, cisPt, and anti-MUC1 when used alone, and PtPz4 and cisPt combined with mAb significantly stimulated *Bax* protein expression (Figure 8A). As shown in Figure 8B, *Bad* mRNA increased as a result of PtPz4 action, and this expression was enhanced by anti-MUC1 addition. Significant stimulation of *Bad* mRNA was also observed after PtPz6 combined with anti-MUC1. Bid mRNA expression was significantly inhibited by PtPz6 when used alone and stimulated by PtPz4 combined with anti-MUC1 (Figure 8C). PtPz6 and cisPt in monotherapy and PtPz6 together with mAb significantly stimulated *Bim* mRNA expression (Figure 8D). However, the addition of anti-MUC1 to PtPz6 did not enhance the stimulatory effect in comparison with PtPz6 monotherapy.

### 3.8. The Effect of PtPz4, PtPz6, cisPt, and Anti-MUC1 on Anti-Apoptotic Factors

*Bcl-xL* and *Bcl-2* are considered crucial pro-survival proteins. To analyze the effect of cisPt, its derivatives, and anti-MUC1 on *Bcl-xL*, RT-PCR and Western blotting were applied. We found that all the examined compounds significantly inhibited *Bcl-xL* mRNA expression, and the effect was enhanced after combining PtPz4 and PtPz6 with anti-MUC1 in comparison with monotherapy (Figure 9A). Such enhancement was not observed after joining cisPt with anti-MUC1. Significant suppression of *Bcl-xL* protein was observed only after PtPz6 and PtPz4 combined with anti-MUC1. We revealed that *Bcl-2* mRNA was significantly inhibited by cisPt, anti-MUC1, and PtPz4 and PtPz6 and cisPt in combination with mAb (Figure 9B). A large stimulation of such inhibition after anti-MUC1 addition was observed.

### 3.9. The Effect of PtPz4, PtPz6, cisPt, and Anti-MUC1 on Caspases

Expression of initiatory *caspase-9* and executive *caspase-3* were assessed in our study. The results presented in Figure 10A revealed that PtPz4, PtPz6, cisPt, and anti-MUC1 significantly increased *caspase-9* mRNA expression in comparison to untreated controls. Such results were also observed for PtPz4 and PtPz6 combined with mAb; however, without the stimulatory effect of anti-MUC1 in comparison with monotherapy. Surprisingly, such stimulatory effect was not observed as an effect of anti-MUC1 combined with cisPt and its derivatives on *caspase-9* protein expression (Figure 10B). Anti-MUC1 alone and combined with PtPz4, PtPz6, and cisPt significantly inhibited pro-*caspase-9* expression. Pro-*caspase-9* was stimulated only by PtPz4 monotherapy. Cleavage of *caspase-9* was suppressed by both cisPt derivatives, anti-MUC1 and mAb combined with PtPz4. There was no effect of cisPt, anti-MUC1 combined with PtPz6 and cisPt on cleaved *caspase-9*. *Caspase-3* mRNA expression significantly increased as a result of anti-MUC1 action and after combined treatment of mAb with cisPt and its derivatives (Figure 10C). Pro-*caspase-3* expression significantly increased after monotherapy with PtPz6, cisPt, and anti-MUC1 compared to control (Figure 10D). Such effect was not observed after PtPz4 used alone. The addition of mAb to cisPt and its derivatives enhanced the noticed effect. All the examined compounds stimulated cleaved *caspase-3* expression with intensified outcome after combined therapy.

### 3.10. The Effect of PtPz4, PtPz6, cisPt, and Anti-MUC1 on Cancer Related Carbohydrate Antigens Expression

Finally, using ELISA tests, we checked the expression of cancer-related sugar antigens Tn, T, sialyl Tn, and sialyl T in cell lysates and culture medium after incubation of cancer cells with examined drugs. Reaction with VVA lectin allowed the dectection of the Tn antigen. We found that both cisPt derivatives as well as anti-MUC1, PtPz4, PtPz6, and cisPt combined with mAb significantly decreased Tn antigen expression in culture medium compared to untreated controls (Figure 11A). All the drugs did not affect the expression of this antigen in cell lysates. T antigen, detected after reaction with PNA lectin was significantly reduced in cell lysates by PtPz4 and anti-MUC1 action as well as by combined therapy of mAb with PtPz4, PtPz6, and cisPt (Figure 11B). CisPt induced this antigen expression in culture medium, and PtPz6, anti-MUC1 as well as PtPz4, PtPz6, and cisPt combined with anti-MUC1 reduced T antigen expression. The inhibitory effect was intensified after combined treatment. Sialylated form of Tn antigen expression was determined using SNA lectin. We revealed inhibition of sialyl Tn antigen expression in cell lysates after combined action of anti-MUC1 with PtPz4 and PtPz6 (Figure 11C). Monotherapy with cisPt and anti-MUC1 inhibited this antigen expression in the culture medium compared to controls. The addition of anti-MUC1 to cisPt and its derivatives markedly intensified such an inhibitory effect. MAAII lectin was used to detect sialyl T antigen expression. We found that this antigen expression in cell lysates was suppressed only by combined treatment of anti-MUC1 and PtPz6 (Figure 11D). In culture medium, a significant decrease in sialyl T antigen was revealed after PtPz4, anti-MUC1 monotherapy, as well as a result of the combined action of the drugs with anti-MUC1.

## 4. Discussion

Cisplatin is one of the best and first metal-based chemotherapeutics [22,23]. It is active against a variety of solid cancers such as testicular, ovarian, bladder, lung, cervical, gastric, head and neck cancer, and some others [4,6]. However, due to severe side effects and acquired drug resistance, its clinical application can be limited. Therefore, much effort has been put into searching for new cisplatin-based anti-cancer complexes [6]. Recently, new pyrazole-platinum(II) complexes have been applied as promising anti-cancer agents leading to cell cycle arrest and apoptosis in MCF-7 and MDA-MB-231 breast cancer cells [7]. Moreover, combined therapies of cisplatin or its complexes with other drugs have been revealed to acquire better effectiveness, conquer drug resistance and lower undesirable toxicity [4].

As was mentioned in the Introduction section, tumor-associated MUC1 plays a crucial role in the development of many cancers. It has been reported to inhibit cell death and promote cell differentiation, proliferation, invasiveness, and metastasis [8]. Thus, inhibition of MUC1 activities could likely be an advantageous strategy in cancer therapy [24]. Moreover, its high expression, differential distribution pattern relative to normal cells, and altered architecture in malignant tissues made this molecule a top molecular target in anti-cancer treatment [13,14,25,26,27]. Antibodies have been confirmed to be compelling additions to therapy for many types of cancer [28]. A number of anti-MUC1 monoclonal antibodies have been described in the literature, and some of them have been involved in pre-clinical studies [13,24,29,30]. In this study, we demonstrated the effects of combined action of an anti-MUC1 monoclonal antibody with cisplatin and two pyrazole-platinum complexes on apoptosis and cell death-related factors in CRL-1739 gastric cancer cells. We applied IgG class anti-MUC1 (BC2) monoclonal antibody reacting with a five amino acid epitope of the MUC1 core protein with low susceptibility for glycosylation.

Hisatsune et al. [31] reported that MUC1 mucin on the cell surface could be internalized by anti-MUC1 antibody through the macropinocytic pathway. It has been also demonstrated that antibodies, by direct binding to their receptors, are able to inhibit their activity, resulting in the inhibition of signaling cascades that promote cell cycle and function [28]. In our research, we showed that 24 h incubation of AGS cancer cells with anti-MUC1 monoclonal antibody resulted in reduced expression of *MUC1* gene as well as mucin expression in cell lysates and culture medium. Upon such results, we assumed that anti-MUC1 mAb potentially inhibited the activity and function of MUC1. Such a conclusion was supported by revealing an inhibitory effect of mAb on the MUC1 cytoplasmic tail. This part of MUC1 was reported to be involved in intracellular signal transduction through the interactions with several signaling molecules implicated in the cancer regulation by affecting the proliferation, apoptosis, and transcription of various genes [32,33]. Thus, we can speculate that revealed in our study, MUC1 mRNA inhibition triggered by anti-MUC1 action could occur via the MUC1 cytoplasmic tail. Combined administration of anti-MUC1 and two cisPt derivatives only slightly enhanced the inhibitory effect towards this part of MUC1. Surprisingly, cisplatin used alone and combined with mAb stimulated MUC1 cytoplasmic tail expression.

Cisplatin complexes used in our study were PtPz4 [Pt_2_(pyrazole)_4_(berenil)_2_]·4HCl·2H_2_O and PtPz6 [Pt_2_(N-methylpyrazole)_4_(berenil)_2_]·4HCl·2H_2_O (Figure 1), the first one with an unsubstituted pyrazole ring, and the second with an ethyl group. These two complexes, as well as four others, with different substituents, were applied by Czarnomysy et al. [7] as anti-cancer agents in MCF-7 and MDA-MB-231 breast cancer cells. They revealed that the activity of the examined complexes depends on the type of pyrazole ligand. It is said that the general action of cisplatin and its complexes are based on interacting with genetic material what in consequence leads to apoptosis [2,3,7]. In our study, we revealed that PtPz6 administrated with anti-MUC1 the most strongly induced apoptosis, as compared with controls, cisPt, anti-MUC1 as well as with both cisPt complexes used alone and PtPz4 combined with mAb. Such a result can suggest that anti-MUC1 induced pro-apoptotic activity of cisPt complex with pyrazole substituted with an ethyl group.

Evasion from apoptosis seems to be one of the most important hallmarks of cancer. NF-κB and *Akt* signaling pathways are among the key factors regulating the process of programmed cell death [24,34]. Thus, blocking of such factors activation is considered to be a highly rational direction in cancer therapy. It has been stated that NF-κB, which is generally regarded as a pro-survival agent, is one of the transcription factors which can be deregulated in gastric cancer. Many of the genes transcribed by NF-κB promote carcinogenesis [35]. However, the correlation between its activity and clinicopathological features is still not fully clarified. It has been also demonstrated that chemotherapy itself could evoke NF-κB activation in gastric cell lines leading to the achievement of chemoresistance [34]. Our results, in relation to NF-κB mRNA expression in AGS cancer cells, support the rationality of cisPt usage, as well as PtPz6 and anti-MUC1, as the factor was suppressed by such treatment. In human cancers, *Akt* is considered a significant mediator of the cell cycle, promoting cancer cell proliferation as well as contributing to resistance to apoptosis [36]. *Akt* action is based on the regulation of activity of several components participating in cell death, such as pro-apoptotic *Bad*, *Bax*, *Bim*, or *caspase-9*. Phosphorylation of *Bad* induced by *Akt* inhibits its heterodimerization with pro-survival *Bcl-xL* leading to restoration of *Bcl-xL* anti-apoptotic function [37]. Therefore, as this factor is usually highly activated in malignant cells, the blocking of its expression seems to be a rational target in cancer therapy. Moreover, numerous studies have clearly reported that activation of the *Akt* signaling pathway could regulate *Bim* expression. Following pro-apoptotic events, *Bim* translocates to the mitochondria, where it is essential to mediate the release of cytochrome c, which in turn activates *caspase-9* [38]. Additionally, *Bim* upregulation has been reported to affect the chemotherapy response [39]. Therefore, the combined action of PtPz6 and anti-MUC1 showing inhibitory effect toward *Akt* mRNA revealed in our study seems to be an effective therapy, promoting activation of apoptosis. Such conclusion can be supported by our outcomes revealing that in the case of PtPz6 administrated with mAb, inhibition of *Akt* gene correlated with *Bad* and *Bim* mRNA stimulation as well as *Bcl-xL* mRNA inhibition. *Bax* is the next pro-apoptotic member regulating apoptosis. Its reduction plays a key role in tumorigenesis. It promotes cell death mainly through mitochondrial membrane permeabilization. On the contrary, pro-survival *Bcl-2* impedes apoptosis by suppressing the activity of *Bax* [40,41]. There have been reports considering *Bax/Bcl-2* ratio as a factor determining cell susceptibility to apoptosis. The authors suggested that a lower level of such a ratio may lead to resistance of cancer cells to apoptosis [42]. Our results do not support the presented idea. We revealed no significant stimulatory effect of applied treatment of *Bax* mRNA. Surprisingly PtPz6, cisPt, and anti-MUC1 monotherapy, as well as combined treatment of PtPz4 and cisPt with mAb, significantly stimulated *Bax* on protein level. We suggest that such lack of correlation between *Bax* mRNA and protein can be explained, e.g., by negative feedback—a high level of protein can downregulate mRNA expression. However, the inhibitory effect of monotherapy with cisPt and mAb as well as anti-MUC1 combined with PtPz4, PtPz6, and cisPt on *Bcl-2* mRNA support rationality of applied therapy in anti-cancer treatment. Pro-apoptotic Bid has been reported as one more factor with a major role in gastric cancer development [43]. Our outcome, revealing stimulation of Bid mRNA expression by PtPz4 administrated with anti-MUC1 gives one more proof of rationality of anti-MUC1 application.

Caspases are a family of proteases that play essential roles in the initiation (e.g., *caspase-8,-9*) and execution (e.g., *caspase-3*) of apoptosis [44]. Thus, it is generally stated that they promote cell death, and their loss is connected to cancer development. However, it was documented that caspases can also be involved in tumorigenesis through a non-apoptotic pathway by influencing proliferation, invasion, and migration [45,46]. Commonly, the initiatory caspase zymogens are transformed into the active proteases by dimerization-induced conformational changes leading to the formation of active, cleaved forms. On the contrary, executioner caspases require cleavage by initiatory caspases for their activation [47]. Upon the results of the presented work, we observed that combining anti-MUC1 with both cisPt derivatives induced apoptosis by enhancement the expression of *caspase-9* and *-3* mRNAs. Such effect was also observed for the action of cisPt and mAb towards *casp-3*, but not *casp-9*. Surprisingly *caspase-9* mRNA expression did not correlate with *caspase-9* protein. We can assume that *caspase-9* underwent such posttranslational modifications, which led to the decreased expression of the protein. We may also suggest too low sensitivity of Western blotting, below that of RT-PCR. Nevertheless, the expression of both forms of *caspase-3* protein, pro- and cleaved, increased with higher intensity after combined therapy. Our result, revealing increased *caspase-3* expression after applied therapy in comparison to untreated control, points to leading cancer cells to apoptosis. Interestingly, Huang et al. [48] have reported a higher level of *caspase-3* protein in untreated malignant, comparing to nonmalignant breast tissue. It was explained by the activation of *caspase-3* in dying cancer cells what resulted in tumor cell repopulation and further growth.

On many epithelial-origin cancer cells, MUC1 is enriched by specific tumor-associated carbohydrate antigens (TACAs), including Tn, T, sTn, and sT sugar structures. They are overexpressed on the surface of cancer cells compared to those on normal tissues [49]. It has been reported that the presence of such truncated glycan structures plays a key role in tumor initiation, progression, and metastasis as TACAs are able to interact with many factors participating in signaling pathways promoting cancer development (leading to the creation of a pro-tumor microenvironment, favoring tumor progression and metastasis) [50]. TACAs expression levels have been used as biomarkers of poor prognosis [51]. Moreover, they have been recently proposed as a specific “glycol-code” that could be considered as a novel immune checkpoint, offering new immunotherapeutic opportunities [49,50]. In our study, we assumed that anti-MUC1 based therapy could also block MUC1 associated TACAs expression. Thus, our results revealing significant inhibition of Tn, T antigens, and their sialylated forms, enhanced by combined therapy, strongly support the rationality of potential anti-MUC1 application in gastric cancer treatment.

## 5. Conclusions

Summarizing our results, we suggest the application of two pyrazole-berenil cisPt complexes combined with anti-MUC1 monoclonal antibody as a promising strategy in gastric cancer treatment, more effective than monotherapy. Generally, the anti-MUC1 addition to cisplatin derivatives enhanced pro-apoptotic effects towards *Bax, Bad, Bcl-xL Bcl-2, caspases*, and TACAs in comparison to untreated control and monotherapy. We realize that not all our results seem to be consistent. Some of them need to be elucidated in the next experiments, planned to be continued in the subject. We are also aware of a limitation of the presented study concerning the usage of only one cancer cell line. In the future, we are going to expand our research by applying at least one more gastric cancer cell line.

## Figures and Tables

**Figure 1 pharmaceutics-13-00968-f001:**
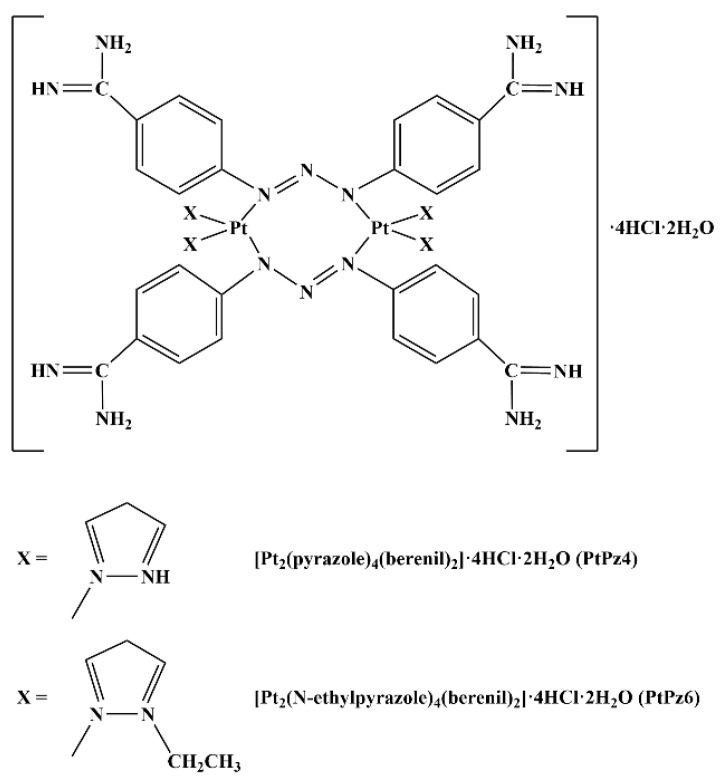
Structures of pyrazole-platinum(II) complexes PtPz4 and PtPz6.

**Figure 2 pharmaceutics-13-00968-f002:**
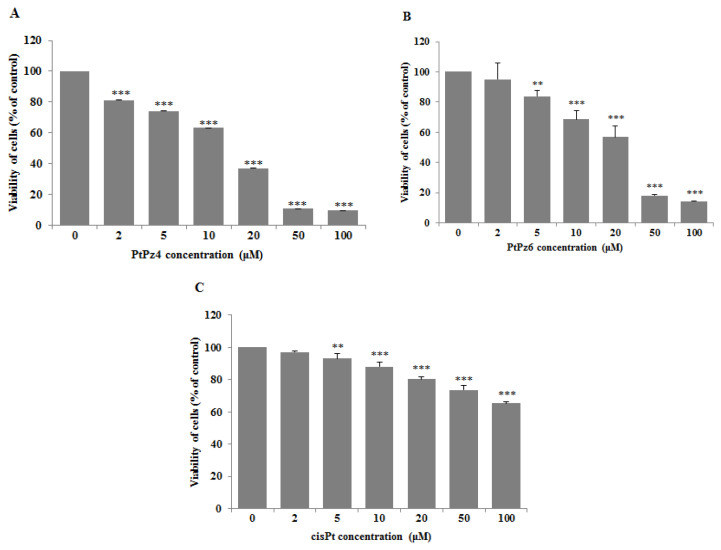
Viability of AGS gastric cancer cells treated for 24 h with 0–100 μM of PtPz4 (**A**), PtPz6 (**B**) and cisPt (**C**). Mean values **±** SD are the mean of triplicate culture. ** *p* < 0.01; *** *p* < 0.001.

**Figure 3 pharmaceutics-13-00968-f003:**
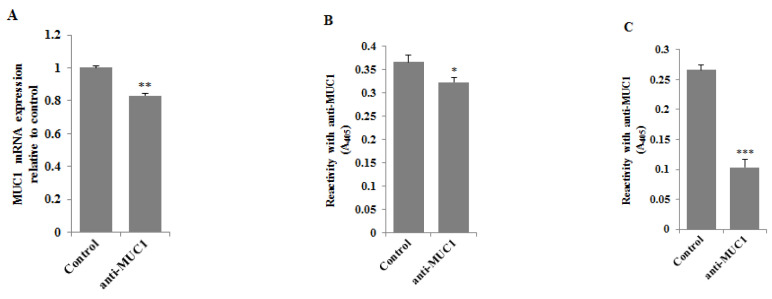
The effect of anti-MUC1 mAb on MUC1 mRNA (**A**), MUC1 glycoprotein expression in cell lysates (**B**), and culture medium (**C**). The AGS gastric cancer cells were incubated for 24 h with anti-MUC1 (5 μg/mL). mRNA was assessed by RT-PCR. The result is shown as a relative fold change in mRNA expression of the gene in comparison to the gene in controls where expression was set at 1 ± SD are the mean of triplicate cultures. ** *p* < 0.01. MUC1 glycoprotein expression was determined by ELISA tests. The results are expressed as absorbance at 405 nm after reactivity with anti-MUC1 monoclonal antibody (BC2 clone). Values ± SD are the mean from three independent assays. * *p* < 0.05; *** *p* < 0.001.

**Figure 4 pharmaceutics-13-00968-f004:**
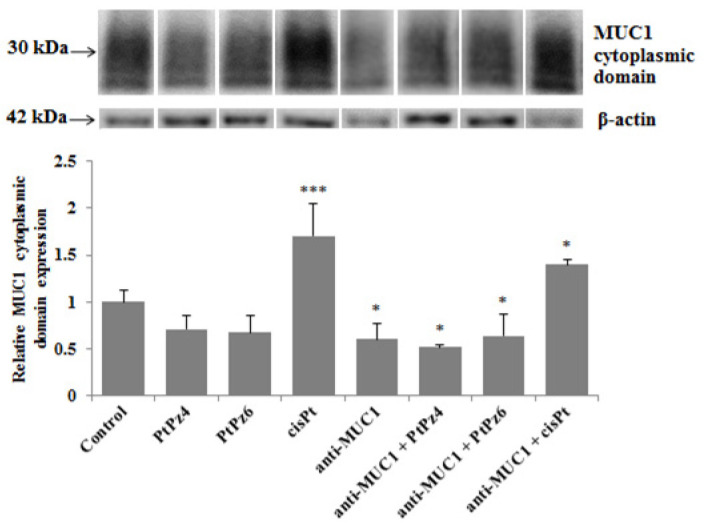
The effect of PtPz4, PtPz6, cisPt, and anti-MUC1 on MUC1 cytoplasmic domain expression in AGS gastric cell lysates determined by Western blotting (with the usage of CT2 clone of anti-MUC1 mAb). The cells were incubated for 24 h with PtPz4 (10 μM), PtPz6 (10 μM), cisPt (10 μM), anti-MUC1 (5 μg/mL), PtPz4 + anti-MUC1 (10 μM + 5 μg/mL), PtPz6 + anti-MUC1 (10 μM + 5 μg/mL), and cisPt + anti-MUC1 (10 μM + 5 μg/mL). The intensities of the bands were quantified by densitometric analysis. Data show the mean ± SD of the relative expression levels (from three assays) standardized to β-actin. * *p* < 0.05; *** *p* < 0.001.

**Figure 5 pharmaceutics-13-00968-f005:**
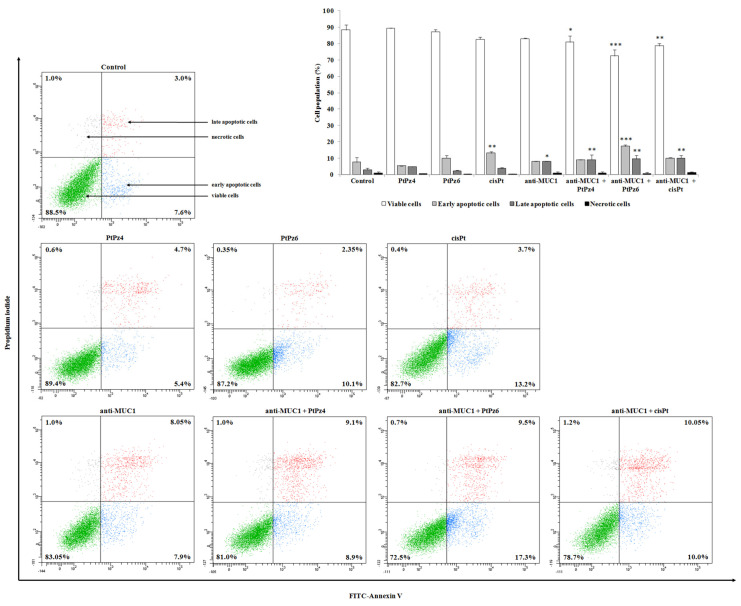
Flow cytometry analysis of AGS gastric cancer cells after 24 h incubation with PtPz4 (10 μM), PtPz6 (10 μM), cisPt (10 μM), anti-MUC1 (5 μg/mL), PtPz4 + anti-MUC1 (10 μM + 5 μg/mL), PtPz6 + anti-MUC1 (10 μM + 5 μg/mL), and cisPt + anti-MUC1 (10 μM + 5 μg/mL) and successive staining with Annexin V and propidium iodide. The data are presented as mean percentage values from three independent experiments (*n* = 3) performed in duplicate. * *p* < 0.05; ** *p* < 0.01; *** *p* < 0.001 versus control group.

**Figure 6 pharmaceutics-13-00968-f006:**
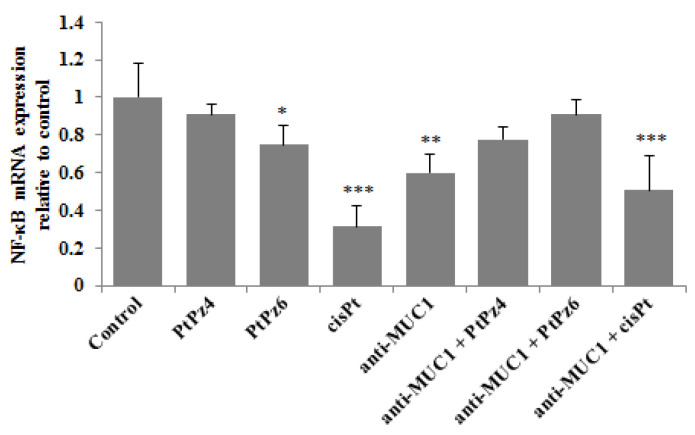
The effect of PtPz4, PtPz6, cisPt, and anti-MUC1 on NF-κB mRNA expression in AGS gastric cell lysates assessed by RT-PCR. The cells were incubated for 24 h with PtPz4 (10 μM), PtPz6 (10 μM), cisPt (10 μM), anti-MUC1 (5 μg/mL), PtPz4 + anti-MUC1 (10 μM + 5 μg/mL), PtPz6 + anti-MUC1 (10 μM + 5 μg/mL), and cisPt + anti-MUC1 (10 μM + 5 μg/mL). The results are seen as a relative fold change in mRNA expression of gene in comparison of gene in control where expression was set at 1 ± SD are the mean of triplicate cultures. * *p* < 0.05; ** *p* < 0.01; *** *p* < 0.001.

**Figure 7 pharmaceutics-13-00968-f007:**
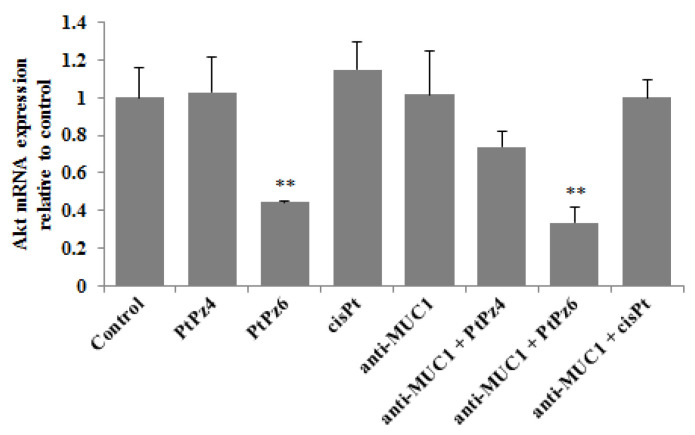
The effect of PtPz4, PtPz6, cisPt, and anti-MUC1 on *Akt* mRNA expression in AGS gastric cell lysates determined by RT-PCR. The cells were incubated for 24 h with PtPz4 (10 μM), PtPz6 (10 μM), cisPt (10 μM), anti-MUC1 (5 μg/mL), PtPz4 + anti-MUC1 (10 μM + 5 μg/mL), PtPz6 + anti-MUC1 (10 μM + 5 μg/mL), and cisPt + anti-MUC1 (10 μM + 5 μg/mL). The results are seen as a relative fold change in mRNA gene expression in comparison of gene in control where expression was set at 1 ± SD are the mean of triplicate cultures. ** *p* < 0.01.

**Figure 8 pharmaceutics-13-00968-f008:**
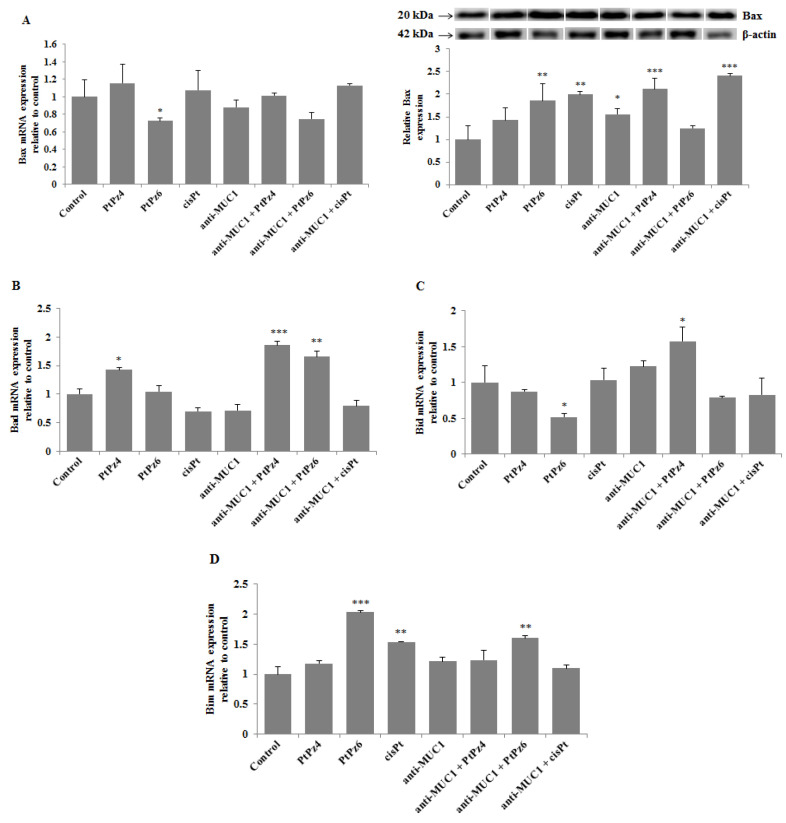
The effect of PtPz4, PtPz6, cisPt, and anti-MUC1 on pro-apoptotic *Bax* mRNA, *Bax* protein (**A**), *Bad* mRNA (**B**), Bid mRNA (**C**), and *Bim* mRNA (**D**) expressions. The cells were incubated for 24 h with PtPz4 (10 μM), PtPz6 (10 μM), cisPt (10 μM), anti-MUC1 (5 μg/mL), PtPz4 + anti-MUC1 (10 μM + 5 μg/mL), PtPz6 + anti-MUC1 (10 μM + 5 μg/mL), and cisPt + anti-MUC1 (10 μM + 5 μg/mL). mRNAs were assessed by RT-PCR. The result is shown as a relative fold change in mRNA gene expression in comparison of gene in control where expression was set at 1 ± SD are the mean of triplicate cultures. * *p* < 0.05; ** *p* < 0.01; *** *p* < 0.001. Expression of *Bax* protein in cell lysates was determined by Western blotting. The intensities of the bands were quantified by densitometric analysis. Data represent the mean ± SD of the relative expression levels (from three assays) normalized to β-actin. * *p* < 0.05; ** *p* < 0.01; *** *p* < 0.001.

**Figure 9 pharmaceutics-13-00968-f009:**
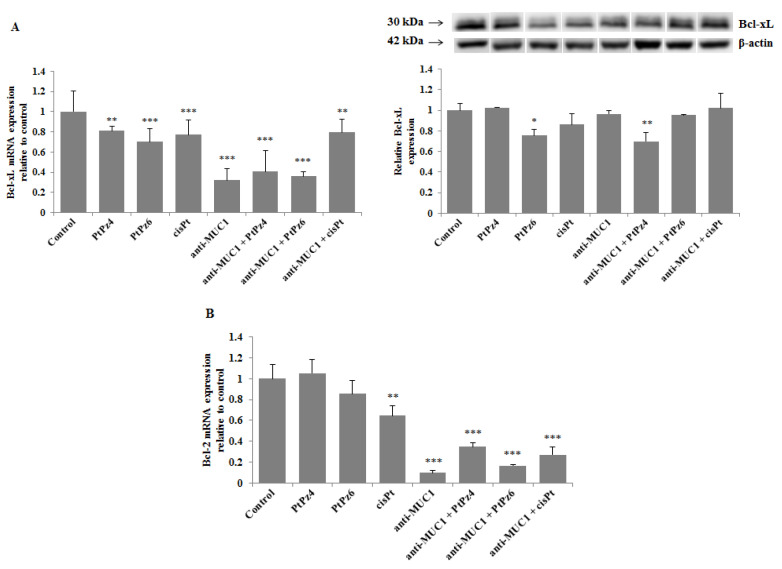
The effect of PtPz4, PtPz6, cisPt, and anti-MUC1 on anti-apoptotic *Bcl-xL* mRNA, *Bcl-xL* protein (**A**), and *Bcl-2* mRNA (**B**) expressions. The cells were incubated for 24 h with PtPz4 (10 μM), PtPz6 (10 μM), cisPt (10 μM), anti-MUC1 (5 μg/mL), PtPz4 + anti-MUC1 (10 μM + 5 μg/mL), PtPz6 + anti-MUC1 (10 μM + 5 μg/mL), and cisPt + anti-MUC1 (10 μM + 5 μg/mL). mRNAs were assessed by RT-PCR. The result is presented as a relative fold change in mRNA gene expression in comparison of gene in control (where expression was set at 1) ± SD are the mean of triplicate cultures. ** *p* < 0.01; *** *p* < 0.001. The expression of *Bcl-xL* protein in cell lysates was determined by Western blotting. The intensities of the bands were quantified by densitometric analysis. Data represent the mean ± SD of the relative expression levels (from three assays) normalized to β-actin. * *p* < 0.05; ** *p* < 0.01.

**Figure 10 pharmaceutics-13-00968-f010:**
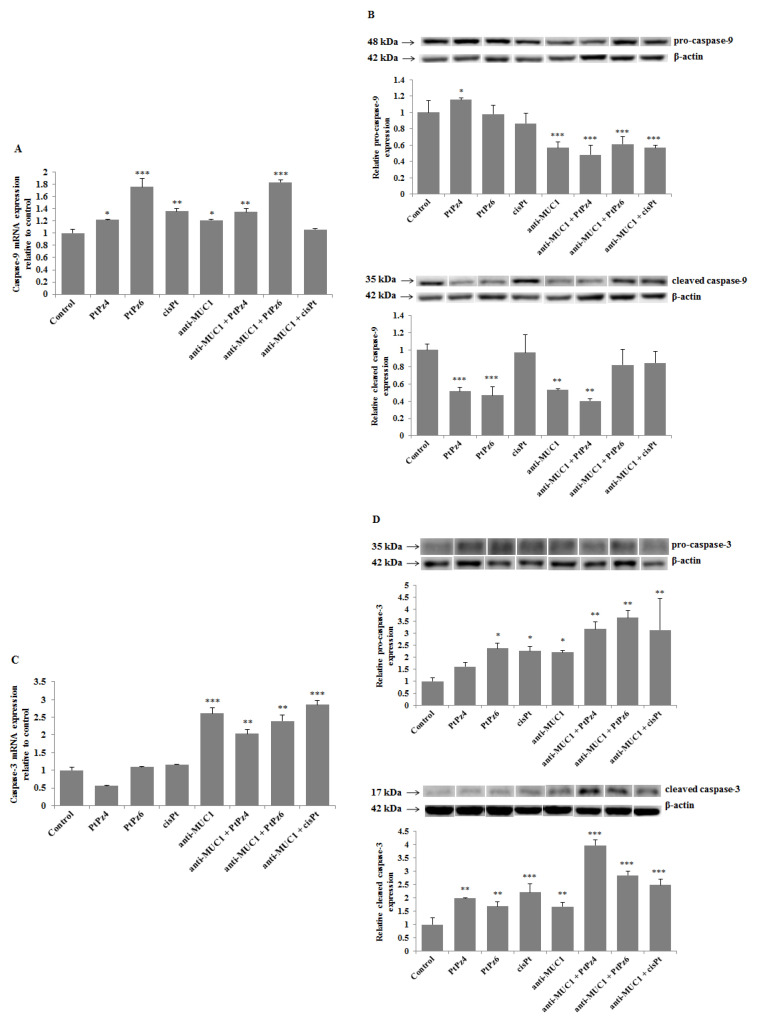
The effect of PtPz4, PtPz6, cisPt, and anti-MUC1 on *caspase-9* mRNA (**A**), pro- and cleaved *caspase-9* (**B**), *caspase-3* mRNA (**C**), and pro- and cleaved *caspase-3* (**D**) expressions. The cells were incubated for 24 h with PtPz4 (10 μM), PtPz6 (10 μM), cisPt (10 μM), anti-MUC1 (5 μg/mL), PtPz4 + anti-MUC1 (10 μM + 5 μg/mL), PtPz6 + anti-MUC1 (10 μM + 5 μg/mL), and cisPt + anti-MUC1 (10 μM + 5 μg/mL). mRNAs were assessed by RT-PCR. The result is presented as a relative fold change in mRNA gene expression in comparison of gene in control where expression was set at 1 ± SD are the mean of triplicate cultures. * *p* < 0.05; ** *p* < 0.01; *** *p* < 0.001. Pro- and cleaved caspase expressions in cell lysates were assessed by Western blotting. The intensities of the bands were quantified by densitometric analysis. Data represent the mean ± SD of the relative expression levels (from three assays) normalized to β-actin. * *p* < 0.05; ** *p* < 0.01; *** *p* < 0.001.

**Figure 11 pharmaceutics-13-00968-f011:**
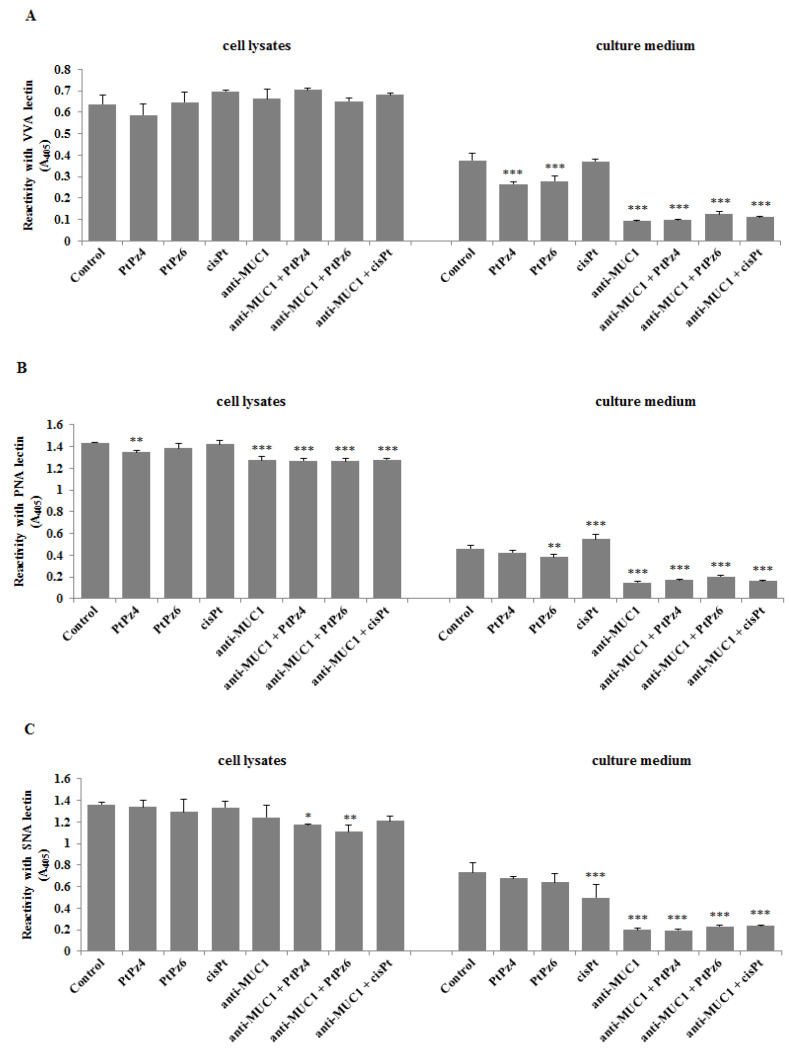
The effect of PtPz4, PtPz6, cisPt, and anti-MUC1 on cancer related Tn antigen (detected by VVA lectin) (**A**), T antigen (detected by PNA lectin) (**B**), sialyl Tn (detected by SNA lectin) (**C**), and sialyl T (detected by MAAII lectin) (**D**). The cells were incubated for 24 h with PtPz4 (10 μM), PtPz6 (10 μM), cisPt (10 μM), anti-MUC1 (5 μg/mL), PtPz4 + anti-MUC1 (10 μM + 5 μg/mL), PtPz6 + anti-MUC1 (10 μM + 5 μg/mL), and cisPt + anti-MUC1 (10 μM + 5 μg/mL). The relative expressions of antigens in cell lysates and culture medium were assessed by ELISA tests. The results are expressed as absorbance at 405 nm after reactivity with biotinylated lectins. Values ± SD are the mean from three independent assays. * *p* < 0.05; ** *p* < 0.01; *** *p* < 0.001.

**Table 1 pharmaceutics-13-00968-t001:** Sequences of primers used for real-time PCR.

Gene	Forward Primer (5′→ 3′)	Reverse Primer (5′→ 3′)
MUC1	TGCCTTGGCTGTCTGTCAGT	GTAGGTATCCCGGGCTGGAA
*Akt*	TCTATGGCGCTGAGATTGTG	CTTAATGTGCCCGTCCTTGT
NF-κB	CTGAACCAGGGCATACCTGT	GAGAAGTCCATGTCCGCAAT
*Caspase-3*	CAGTGGAGGCCGACTTCTTG	TGGCACAAAGCGACTGGAT
*Caspase-9*	CCCATATGATCGAGGACATCCA	ACAACTTTGCTGCTTGCCTGTTAG
*Bcl-2*	GCTGAAGATTGATGGGATCG	TACAGCATGATCCTCTGTCAAG
*Bcl-xL*	TGACGTGGACATCCGC	CTGGAAGGTGGACAGCGAGC
*Bid*	CCTACCCTAGAGACATGGAGAAG	TTTCTGGCTAAGCTCCTCACG
*Bad*	CCCAGAGTTTGAGCCGAGTG	CCCATCCCTTCGTCGTCCT
*Bim*	TAGGTGAGCGGGAGGCTAGGGATCA	GTGCAGGCTCGGACAGGTAAAGGC
*Bax*	TTGCTTCAGGGTTTCATCCA	CAGCCTTGAGCACCAGTTTG
GAPDH	GTGAACCATGAGAAGTATGACAA	CATGAGTCCTTCCACGATAC

**Table 2 pharmaceutics-13-00968-t002:** Antibodies used in the study.

Antibody	Clone	Source
Anti-MUC1; extracellular domain (mouse IgG)	BC2	Abcam
Anti-MUC1; cytoplasmic tail (Armenian hamster IgG)	CT2	Abcam
Anti-*Caspase*-*3* (mouse IgG)	B-4	Santa Cruz Biotech
Anti-Cleaved *caspase*-*3* (rabbit IgG)	5A1E	Cell Sign Tech
Anti-*Caspase*-*9* (mouse IgG)	C9	Cell Sign Tech
Anti-Cleaved *caspase*-*9* (rabbit IgG)	E5Z7N	Cell Sign Tech
Anti-*Bax* (rabbit IgG)	D2E11	Cell Sign Tech
Anti-*Bcl*-*x*L (mouse IgG)	7B2.5	Santa Cruz Biotech
Anti-β-actin (rabbit IgG)		Sigma
Anti-mouse IgG peroxidase conjugated		Sigma
Anti-rabbit IgG peroxidase conjugated		Sigma
Anti-Armenian hamster IgG peroxidase conjugated		Abcam

## Data Availability

All relevant data are included in the article.

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
