# Peer review of "Combined Action of Anti-MUC1 Monoclonal Antibody and Pyrazole-Platinum(II) Complexes Reveals Higher Effectiveness towards Apoptotic Response in Comparison with Monotherapy in AGS Gastric Cancer Cells"

_pharmaceutics, 2021, doi:10.3390/pharmaceutics13070968_

Round 1
Reviewer 1 Report
The manuscript has been revised appropriately and it is acceptable for publication in present form.
Author Response
Reviewer accepted the manuscript. The correstions wre not needed
Reviewer 2 Report
Thank you for the work on the manuscript that is improved a lot, only one point that I would like to mention:
Wherever you take the bands from the Western blots from different experiments or after reorganization of your blots, there should be a little gap in between to indicate that something was cut as otherwise you could any time be blamed of having manipulated your data. It may not look that nice to have little gaps but is indeed the valid way to present data rather than trying to hide that data were taken from different blots.
Author Response
Images from Western blotting (Fig. 4, 8A, 9A, 10B, 10D) were corrected according to suggestions.
Round 2
Reviewer 2 Report
thanks for the corrections, manuscript can be published now
This manuscript is a resubmission of an earlier submission. The following is a list of the peer review reports and author responses from that submission.
Round 1
Reviewer 1 Report
Major comments
The authors demonstrate that the combination of pyrazole-platinum (II) complexes with anti-MUC1 monoclonal antibody more effectively inhibits growth of AGS gastric cell line than the single administration of pyrasole-platinum (II) complexes and the theses effects are explained by the enhanced activation of apoptosis. The idea to use anti-MUC1 antibody in combination with platinum complex is not novel, since the block of MUC1 has already been considered as a potential target for breast cancer treatment in vitro (Glycoconj J 2013, 30: 227-236) and also in vivo (Clin Can Res 2009, 15: 100-109). In addition, some results obtained by the combination of anti-Muc1 antibodies with platinum complexes have been reported in breast cancer cell lines as is described by the authors (Refs. 17 &18). The results and discussion are presented appropriately, however, manuscript must be improved in certain points. In particular,
- There are two anti-MUC1 antibodies listed in Table2, however, there is no specification in the legend to Figures. This must be clarified.
- The authors observed significant decrease of MUC1 mRNA (lines 231-232 and Figure 3A). It is expected to see down-regulation of MUC1 protein levels in cell lysates and culture medium, however, it is not clear how the MUC1 mRNA is reduced. Please explain.
- The authors performed experiments on a single gastric cell line, AGS. The conclusion drawn using AGS could be generally applied to other gastric cell lines with abundant expression of MUC1, or other MUCs, such as MUC5B (MKN45, KATOIII, etc.). At least please discuss.
Minor points
- The order of the presentation of Figure 3 must be A, B and C from left to right.
- The quality of the WB image in Figure 4 must be improved.
- Concerning the MTT assay, in general, cells are treated with substances that have to be tested for certain amount of time, and then MTT is added for 2-3 hours before measurement. We never treat cells with any reagent in the presence of MTT.
- “m” is missing on lien15 and line 194.
Reviewer 2 Report
Supriunik et al. show in their manuscript “Combined action of anti-MUC1 monoclonal antibody and pyrazole-platinum(II) complexes reveals higher effectiveness towards apoptotic response in comparison with monotherapy in AGS gastric cancer cells” that a the tested platin-complexes in combination with anti-Mucin1 are more effective than platin alone. Thereby they use various techniques like proliferation and apoptosis determination, real-time PCR, ELISA and Western blotting.
It is in principle an interesting study showing the potential of a platin-based therapy combined with a Mucin-1 antibody in gastric cancer but it can not be published in the current form as data quality needs to be improved.
Major issues:
- The platin-derivatives were dissolved in DMSO à what was the stock concentration of the derivatives? A dilution (what is the dilution factor?) with media to 10 µM and then going for an analysis of up to 100 µM for viability analysis does not make any sense. Any DMSO concentration around 0.5 to 1% already induced apoptosis itself and does not show any specific effect of the platin-derivatives.
- A control of medium only is one control, but another important control – DMSO in the highest dose used for the platin-derivatives is missing for all experiments. How can you rule out that toxic as well as signaling-induced effects come from the platin-derivatives/a-MUC1 antibody and are no effect of general DMSO toxicity?
- Western blotting: beta-actin controls are a nightmare, they show either no even loading or that cells with the respective treatment are already died too much. Whole protein loading (like Ponceau staining) should be shown to ensure equal loading for the experiments as beta-actin does not show this and therefore, data are not reliable. Suppose that loading would be equal, but beta-actin is not, it is no good reference protein and a better one would have to be found.
- For several Western blots (e.g. Fig. 4, 9 10), controls are not normalized to 1, but to whatever value. If you normalize expression, it should be something like 1 and nothing random.
- How is it possible that in Fig 10 DMSO control shows such a strong pro-Caspase 9 cleavage in the control state? Healthy cells should not die, so there must be a severe problem in the experiment.
- If it is a problem that cells are dying too much after 24 h, it should be considered to do the analysis after 12 h, 16 h or 20 h.
Minor:
- For all axes: whenever using decimals, it should be with dot and not with comma, it is according to English system.
- Line 194: typo: most probably 100 µL?
- Line 203: plusminus sign is missing